# Enhancement of Oral Mucosal Regeneration Using Human Exosomal Therapy in SD Rats

**DOI:** 10.3390/biomedicines13071785

**Published:** 2025-07-21

**Authors:** Chien Ming Lee, Qasim Hussain, Kuo Pin Chuang, Hoang Minh

**Affiliations:** 1Kao-Yi Animal Hospital, Chiayi 600, Taiwan; chase6383396@gmail.com; 2Department of Tropical Agriculture and International Cooperation, National Pingtung University of Science and Technology, Pingtung 912, Taiwan; m11322352@st.npust.edu.tw; 3International Degree Program in Animal Vaccine Technology, International College, National Pingtung University of Science and Technology, Neipu Township, Pingtung 912, Taiwan; 4Graduate Institute of Animal Vaccine Technology, College of Veterinary Medicine, National Pingtung University of Science and Technology, Neipu Township, Pingtung 912, Taiwan; 5School of Medicine, Kaohsiung Medical University, Kaohsiung 807, Taiwan; 6Companion Animal Research Center, National Pingtung University of Science and Technology, Neipu Township, Pingtung 912, Taiwan; 7Department of Anatomy and Histology, Faculty of Veterinary Medicine, Vietnam National University of Agriculture, Hanoi 100000, Vietnam

**Keywords:** human exosomes, oral tissue regeneration, epithelial development, inflammatory regulation, collagen architecture, SD rats, wound healing, regenerative therapeutics

## Abstract

**Background/Objectives:** Oral cavity wound recovery presents unique challenges due to constant moisture exposure and functional mechanical stresses. Nanoscale extracellular vesicles (exosomes) with regenerative properties offer promising therapeutic potential for tissue regeneration, contributing to improved health outcomes. This study evaluated human exosomal preparations in promoting oral mucosal regeneration. **Methods:** We established standardized full-thickness wounds in the buccal mucosa of SD rats and divided subjects into experimental (receiving 50 billion human exosomes) and control (receiving carrier solution only) groups. Comprehensive wound assessment occurred at predetermined intervals (days 0, 3, 7, and 10) through photographic documentation, histological examination, and quantitative measurement. **Results:** Exosomal-treated tissues demonstrated statistically significant acceleration in closure rates (*p* < 0.05), achieving 87.3% reduction by day 10 versus 64.1% in the controls. Microscopic analysis revealed superior epithelial development, reduced inflammatory infiltration, and enhanced collagen architectural organization in exosomal-treated specimens. Semi-quantitative evaluation confirmed consistently superior healing metrics in the experimental group across all assessment timepoints. **Conclusions:** These findings demonstrate that human exosome preparations significantly enhance oral mucosal regeneration in SD rats, suggesting potential clinical applications for accelerating recovery following oral surgical procedures.

## 1. Introduction

The regenerative dynamics of oral cavity tissues differ fundamentally from cutaneous healing processes due to several distinctive characteristics. The perpetually moist environment, continuous functional stresses during speech and mastication, and specialized cellular architecture create unique challenges for therapeutic interventions [1]. While oral tissues typically demonstrate accelerated recovery compared to dermal wounds, certain systemic conditions—including metabolic disorders, compromised immunity, and radiation exposure—can significantly impair this regenerative capacity, potentially resulting in persistent lesions. Contemporary approaches for addressing oral tissue injuries include various topical formulations, bioactive agents, and engineered tissue substitutes; however, these interventions frequently yield suboptimal outcomes, highlighting the necessity for innovative therapeutic strategies [2].

Nano-sized extracellular vesicles, known as exosomes, are released by heterogeneous cell populations and are intercellular communication vehicles [3]. Recent research has shown that exosomes are a viable option for tissue regeneration applications due to their demonstrated capacity to influence fundamental healing processes, including cellular proliferation, migration, vascular development, and matrix reorganization [4,5]. Human-derived exosomal preparations have exhibited particularly notable regenerative capabilities across diverse tissue types, including integumentary, osseous, and cardiac structures [6,7].

The application of exosomal preparations for oral tissue regeneration represents an innovative approach with several advantages over conventional therapies. Unlike cellular transplantation approaches, exosomes lack replicative capacity and, consequently, present reduced immunogenicity and tumorigenic potential [8]. Their microscopic dimensions facilitate efficient tissue penetration, and their molecular cargo can be modified through parent cell manipulation or specialized loading techniques [9].

Despite accumulating evidence supporting exosomal efficacy in dermal wound recovery, their potential application in oral mucosal regeneration remains relatively unexplored [10]. Furthermore, optimal administration protocols, delivery methodologies, and application timing for exosomal treatment of oral wounds have yet to be established through systematic research.

The purpose of this study is to use an SD rat model to examine the therapeutic potential of human exosome preparations to promote oral mucosal regeneration. We hypothesized that the application of 50 billion human exosomes would enhance wound closure rates, improve epithelial development, reduce inflammatory responses, and promote organized collagen formation compared to control treatment. By assessing both macroscopic and microscopic healing parameters throughout a 10-day observation period, this study provides comprehensive insights into how human exosome preparations influence oral mucosal regeneration processes.

## 2. Materials and Methods

### 2.1. Experimental Design

This study followed the animal study protocol reviewed and approved by the Institutional Animal Care and Use Committee (IACUC; protocol code: 25PLAN-20; approved on 23 January 2025). Before the experiment, 24 Sprague-Dawley male rats weighing 250–300 g and aged eight weeks were purchased and acclimatized to regular living conditions for seven days. These conditions consisted of a 12 h light/dark cycle, 22 ± 2 °C temperature, and 55 ± 10% humidity. The animals were fed a regular rodent laboratory diet and were given unlimited access to water during the trial.

Random assignment was used to place the animals in the control group receiving carrier solution alone or the treatment group receiving human exosomes.

### 2.2. Exosome Characterization

Human umbilical cord mesenchymal stem cell-derived exosomes (HUSC-E) were manufactured using a standardized clinical protocol involving human umbilical cord mesenchymal stem cell culture at a 1 × 10^8^ cells/mL density, followed by supernatant harvesting, PH and conductivity adjustment, Tangential Flow Filtration (TFF) for 15-fold concentration, ADP incorporation for integrity maintenance, freeze-drying, and gamma irradiation sterilization.


*Size Distribution Analysis:*


The exosome size distribution and concentration were analyzed using nanoparticle tracking analysis (NTA) (NanoSight NS300, Malvern Panalytical, Malvern, Worcestershire, UK). The samples were diluted in PBS and mixed well, and the diluted samples were illuminated by a monochromatic laser beam at 532 nm to register a 60 s video taken with a mean frame rate of 30 frames/s, yielding particle concentrations in the region of 10^8^–10^9^ particles/mL, in accordance with the manufacturer’s recommendations. The modal particle size was 92.7 nm with a particle concentration of 3.49 × 10^11^ particles/mL, confirming the presence of extracellular vesicles within the expected size range for exosomes (30–150 nm) [11] (Figure 1 and Figure 2).

### 2.3. Surgical Methodology and Treatment Administration

Xylazine (10 mg/kg) and ketamine (80 mg/kg) were injected intraperitoneally to provide general anesthesia for all the procedures. Following confirmation of adequate anesthesia depth, the oral cavity was disinfected with 0.12% chlorhexidine solution. Standardized full-thickness wounds (4 mm diameter) were created in the buccal mucosa bilaterally using sterile biopsy punches, with careful attention to avoid damage to underlying neurovascular structures [12]. Hemostasis was achieved through gentle pressure application with sterile gauze.

Immediately following wound creation, treatment was administered according to group assignment. The experimental group received 10 μL of solution containing 50 billion human exosomes applied directly to the wound surface using a micropipette. The control group received an identical volume of carrier solution without exosomal content. All the animals received post-surgery analgesia (subcutaneous buprenorphine 0.05 mg/kg) every 12 h for 48 h, and all the animals were monitored during the trial for any discomfort, infection, or complications.

### 2.4. Macroscopic Assessment

Wound healing progression was documented through standardized digital photography (Canon EOS 90D with macro lens, Tokyo, Japan). Two independent evaluators who were blind to treatment allocation used ImageJ software (version 1.54, NIH, Bethesda, MD, USA) to estimate the extent of the wound [13]. The formula **“[(Original wound area − current wound area)/original wound area] × 100”** was used to determine the percentage of wound closure [14].

### 2.5. Tissue Collection and Processing

Animals were euthanized through carbon dioxide inhalation followed by cervical dislocation. The entire buccal mucosa was carefully removed and then preserved for a day in 10% neutral buffered formalin. Following fixation, specimens underwent normal histology methods, including xylene washing, paraffin embedding, and a series of graded ethanol dehydrations. Serial sections (5 μm thick) were sectioned on a rotary microtome from the mid-wound site and mounted on positively charged glass slides for additional histological study.

### 2.6. Histological Evaluation

For a general morphological assessment, tissue slices were stained with hematoxylin and eosin (H&E), and collagen was observed using Masson’s trichrome. Histological evaluation was performed by an experienced oral pathologist blinded to treatment allocation using a light microscope (Olympus BX53, Tokyo, Japan) equipped with digital imaging capabilities. The following parameters were assessed:**Epithelial regeneration:** measured as the epithelial thickness (μm) at the wound center and margins and evaluated for the completeness of coverage, cellular organization, and stratification;**Inflammatory response:** quantified through inflammatory cell counts (neutrophils, macrophages, and lymphocytes) in the wound bed and margins;**Collagen deposition and organization:** evaluated for density, fiber orientation, and architectural pattern using a semi-quantitative scoring system (1 = minimal/disorganized to 4 = abundant/well organized).

Additionally, a composite histological score was calculated based on the above parameters to provide an overall assessment of healing quality, with higher scores indicating superior regeneration.

### 2.7. Data Analysis

Python (version 3.9) with Matplotlib (version 3.5.1) was used for data analysis and visualization. The distribution normality of the data was examined using the Shapiro–Wilk test. Furthermore, Student’s *t*-test was employed to compare groups in continuous variables, which are represented as the mean ± standard deviation. The time-dependent healing of wounds was assessed using two-way ANOVA with Bonferroni post hoc analysis. All analyses were considered statistically significant when *p* < 0.05. Graphs were generated using Matplotlib’s pyplot interface with customized parameters to ensure clear visualization of experimental outcomes. For macroscopic wound measurements, three technical replicates were performed for each wound, with the mean value used for statistical analysis. Epithelial thickness measurements were conducted at five different locations across each specimen, and the average value was used for analysis. Inflammatory cell counts were performed in triplicate across three non-overlapping high-power fields per specimen. Collagen assessment was conducted independently by two blinded evaluators. All quantitative data are presented as the mean ± standard deviation, with error bars in the figures representing the standard deviation of the biological replicates. All measurements were conducted under tightly controlled conditions using standardized protocols, and blinded evaluators assessed the outcomes to reduce measurement bias.

## 3. Results

### 3.1. Macroscopic Healing Progression

During the observation period, the group treated with human exosomes showed faster wound closure than the control group, according to macroscopic inspection. By day 3 post-wounding, the experimental group demonstrated a significantly greater reduction in the wound area (32.4 ± 5.2% closure) compared to the control group (19.1 ± 4.3% closure, *p* < 0.05). This enhanced healing trajectory continued at day 7, with exosome-treated wounds achieving 68.7 ± 4.9% closure versus 42.3 ± 5.6% in the controls (*p* < 0.01). By the study endpoint (day 10), the experimental group exhibited near-complete wound resolution (87.3 ± 5.8% closure), while the control wounds demonstrated only partial healing (64.1 ± 6.2% closure, *p* < 0.01) (Figure 3).

Visual assessment of the wound characteristics revealed additional qualitative differences between groups. Exosome-treated wounds displayed minimal peripheral erythema, smooth wound margins, and progressive contraction. In contrast, control wounds exhibited more pronounced inflammatory signs, irregular margins, and slower contraction rates (Figure 4).

### 3.2. Histological Observations

#### 3.2.1. Epithelial Regenerative Patterns

Histological examination revealed significant differences in epithelial regeneration between treatment groups. At day 3, exosome-treated specimens demonstrated early epithelial migration from the wound margins (mean thickness 185.3 ± 12.4 μm) compared to minimal epithelial activity in the controls (129.7 ± 10.8 μm, *p* < 0.05). By day 7, the experimental group exhibited substantial epithelial coverage with emerging stratification (345.2 ± 15.6 μm thickness) versus incomplete, thinner epithelium in the controls (240.8 ± 13.2 μm, *p* < 0.01). At day 10, exosome-treated wounds displayed nearly complete re-epithelialization with well-organized stratification and early keratinization (462.5 ± 18.3 μm thickness), while control specimens showed incomplete coverage with irregular organization (289.4 ± 14.7 μm thickness, *p* < 0.01) (Figure 5).

#### 3.2.2. Inflammatory Response Patterns

Inflammatory cell infiltration patterns differed markedly between groups across all timepoints. At day 3, both groups exhibited inflammatory responses; however, the total inflammatory cell count was significantly lower in exosome-treated specimens (58.3 ± 8.2 cells/HPF) compared to the controls (94.7 ± 10.5 cells/HPF, *p* < 0.05). By day 7, inflammation had substantially resolved in the experimental group (32.6 ± 5.4 cells/HPF) while remaining prominent in controls (67.9 ± 7.8 cells/HPF, *p* < 0.01). At day 10, exosome-treated tissues exhibited minimal residual inflammation (15.2 ± 3.6 cells/HPF) versus persistent moderate inflammation in the control specimens (41.5 ± 6.3 cells/HPF, *p* < 0.01) (Figure 6).

#### 3.2.3. Collagen Architectural Patterns

Masson’s trichrome staining revealed significant differences in collagen deposition and organization between groups. At day 3, exosome-treated specimens demonstrated early collagen deposition with initial organizational patterns (score 1.8 ± 0.4) versus minimal, disorganized collagen in the controls (score 1.1 ± 0.3, *p* < 0.05). By day 7, the experimental group exhibited moderate collagen density with emerging fiber orientation (score 2.9 ± 0.5) compared to less abundant, randomly arranged fibers in the controls (score 1.9 ± 0.4, *p* < 0.01). At day 10, exosome-treated wounds displayed a dense, well-organized collagen architecture resembling normal mucosal lamina propria (score 3.7 ± 0.4), while the control specimens showed moderate density with persistent disorganization (score 2.5 ± 0.5, *p* < 0.01) (Figure 7).

### 3.3. Histological Scoring Analysis

Composite histological scoring confirmed consistently superior healing metrics in the exosome-treated group across all assessment timepoints. At day 3, the experimental group achieved significantly higher overall scores (7.2 ± 0.8) compared to the controls (4.5 ± 0.7, *p* < 0.05). This advantage increased at day 7 (experimental: 10.8 ± 1.1; control: 6.7 ± 0.9, *p* < 0.01) and persisted through day 10 (experimental: 13.6 ± 1.2; control: 8.9 ± 1.0, *p* < 0.01) (Figure 8 and Figure 9).

### 3.4. Statistical Analysis

Bonferroni post hoc tests with two-way ANOVA were performed for all parameters (wound closure, epithelial thickness, inflammatory cell count, and collagen density scores) to assess the main effects of treatment, time, and their interaction. All analyses revealed significant main effects for treatment (wound closure: F = 45.23, *p* < 0.001; epithelial thickness: F = 52.17, *p* < 0.001; inflammatory cells: F = 38.92, *p* < 0.001; collagen density: F = 41.68, *p* < 0.001), time (wound closure: F = 78.91, *p* < 0.001; epithelial thickness: F = 89.45, *p* < 0.001; inflammatory cells: F = 67.33, *p* < 0.001; collagen density: F = 72.14, *p* < 0.001), and significant treatment × time interaction effects (wound closure: F = 12.34, *p* < 0.001; epithelial thickness: F = 8.76, *p* < 0.01; inflammatory cells: F = 6.45, *p* < 0.05; collagen density: F = 7.89, *p* < 0.01). Bonferroni post hoc comparisons confirmed statistically significant differences between treatment groups at each timepoint for all measured parameters. For composite histological scores at day 10, an independent samples *t*-test revealed a significant difference between the groups (t = 8.94, df = 6, *p* < 0.001).

## 4. Discussion

This study demonstrates that human exosome preparations significantly enhance oral mucosal regeneration in an SD rat model, as evidenced by accelerated wound closure, superior epithelial development, modulated inflammatory responses, and improved collagen architectural organization. These findings align with emerging evidence supporting exosomal efficacy in tissue regeneration [15] while specifically addressing the unique characteristics of oral mucosal healing.

The accelerated wound closure observed in exosome-treated specimens likely results from multiple complementary mechanisms. Exosomes have been shown to stimulate cellular proliferation and migration through their bioactive cargo [16]. Additionally, their immunomodulatory properties may create a more favorable microenvironment for tissue regeneration by reducing excessive inflammatory responses that can hinder healing progression [17]. The significant differences in closure rates between experimental and control groups throughout the observation period suggest that these effects manifest early in the healing cascade and persist through later regenerative phases.

The enhanced epithelial regeneration demonstrated in exosome-treated specimens represents a particularly valuable finding for oral mucosal applications. Rapid re-establishment of epithelial integrity is essential for restoring barrier function in the challenging oral environment, where constant moisture exposure and mechanical stresses can compromise healing [18]. Previous studies have demonstrated that exosomes activate epithelial stem cell populations and promote keratinocyte migration through multiple signaling pathways in cutaneous wound healing models [19,20], with similar mechanisms likely applicable to oral mucosal tissues. Our observations of increased epithelial thickness, improved stratification, and accelerated keratinization in the experimental group also suggest that human exosomes effectively stimulate these epithelial regenerative mechanisms in oral mucosal tissues.

The modulation of inflammatory responses observed in exosome-treated specimens aligns with established evidence regarding exosomal immunoregulatory functions. The shift from neutrophil-dominated to macrophage-predominant infiltration in the experimental group suggests the acceleration of the transition from acute to proliferative healing phases [21]. This immunomodulatory effect likely contributes to the overall enhanced healing trajectory by limiting collateral tissue damage from prolonged inflammatory activity while promoting timely progression through subsequent regenerative stages. The significantly reduced inflammatory cell counts in exosome-treated specimens at all timepoints further supports the anti-inflammatory potential of these preparations in oral mucosal applications.

The superior collagen architectural organization demonstrated in the experimental group represents another significant advantage for functional tissue regeneration. Oral mucosal healing typically involves substantial extracellular matrix remodeling to restore tissue integrity and mechanical properties [22]. The observed progression from early, organized collagen deposition to mature, well-aligned fiber architecture in exosome-treated specimens suggests that these preparations effectively modulate fibroblast activity and matrix production. This finding has particular relevance for oral mucosal regeneration, where excessive or disorganized collagen deposition can lead to scar formation and functional impairment [23].

The therapeutic efficacy observed in this study likely stems from the diverse molecular cargo carried within hMSC-derived exosomes, including growth factors such as vascular endothelial growth factor (VEGF), platelet-derived growth factor (PDGF), transforming growth factor-β (TGF-β), and hepatocyte growth factor (HGF), which promote angiogenesis, cellular proliferation, and tissue remodeling [24,25]. Additionally, hMSC-exosomes are enriched with regulatory microRNAs including miR-21, miR-146a, miR-223, and miR-125b, which modulate inflammatory responses and enhance wound healing processes [26,27]. These bioactive molecules work through multiple interconnected molecular mechanisms including TLR4/NF-κB pathway modulation for anti-inflammatory effects, TGF-β/Smad signaling regulation for balanced healing, PI3K/AKT and Wnt/β-catenin pathway activation for cellular proliferation, and VEGF/VEGFR2-mediated angiogenesis, as demonstrated in cutaneous wound healing models [28,29], with similar pathways likely operative in oral mucosal tissues. The regenerative effects likely result from exosome interactions with multiple cell types including keratinocytes (enhanced proliferation and migration), fibroblasts (increased collagen synthesis), and immune cells (modulated inflammatory responses). Whether these effects are direct or indirect through reduced inflammation represents an important area for future investigation.

Current evidence demonstrates that exosomes have unique biological properties that significantly reduce immunogenic responses during cross-species administration. Research has specifically investigated this and found that human exosomes administered to rats were successfully incorporated into rat tissues without triggering detectable immune reactions or adverse effects [30]. This lack of immunogenicity can be explained by several structural and functional characteristics of exosomes: their nanoscale dimensions (30–150 nm), protective lipid bilayer composition, and specialized surface proteins that facilitate rapid cellular uptake while avoiding immune system recognition [31]. Furthermore, unlike whole-cell xenotransplantation, which typically triggers robust immune rejection, exosomes largely bypass allo-recognition mechanisms due to their acellular nature and ability to quickly enter target cells before significant immune interaction occurs [32]. A recent study has confirmed these observations, documenting that exosomes maintain good biocompatibility with low toxicity and minimal immunogenicity even when crossing species barriers [33,34]. These properties make human-derived exosomes valuable tools for translational research, enabling the delivery of therapeutic molecules across species without compromising safety or triggering significant immune responses.

Our study has some limitations. First, the relatively short observation period may not capture long-term outcomes or potential delayed effects of exosomal treatment. Second, our assessment methodology did not include molecular or cellular analyses such as immunohistochemistry for phosphorylated proteins or Western blot analysis that might elucidate specific mechanistic pathways underlying the observed regenerative effects. Such mechanistic validation would strengthen the link between exosome treatment and the proposed cellular mechanisms underlying the observed regenerative effects. Future studies should address these limitations.

## 5. Conclusions

This study demonstrates that human exosome preparations significantly enhance oral mucosal regeneration in SD rats across multiple healing parameters, including wound closure rates, epithelial development, inflammatory regulation, and collagen architectural organization. These findings suggest considerable therapeutic potential for exosomal applications in clinical scenarios requiring accelerated or improved oral tissue regeneration, such as following surgical procedures, traumatic injuries, or in patients with compromised healing capacity.

Future research directions should focus on elucidating specific molecular mechanisms underlying these regenerative effects, optimizing delivery methodologies for various oral applications, and evaluating potential synergistic effects with established therapeutic approaches. Additionally, the investigation of dose–response relationships and treatment timing could further refine clinical protocols for maximal efficacy. As exosomal therapeutics continue to advance toward clinical implementation, their application in oral regenerative procedures represents a promising frontier for improving patient outcomes through enhanced tissue restoration.

## Figures and Tables

**Figure 1 biomedicines-13-01785-f001:**
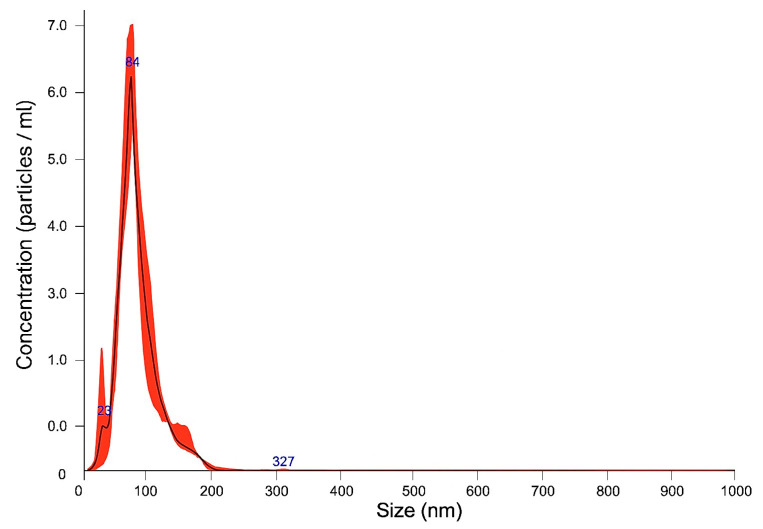
Nanoparticle tracking analysis (NTA) of exosome preparations. Size distribution analysis shows modal particle size of 92.7 nm, confirming presence of extracellular vesicles within the expected exosome size range (30–150 nm).

**Figure 2 biomedicines-13-01785-f002:**
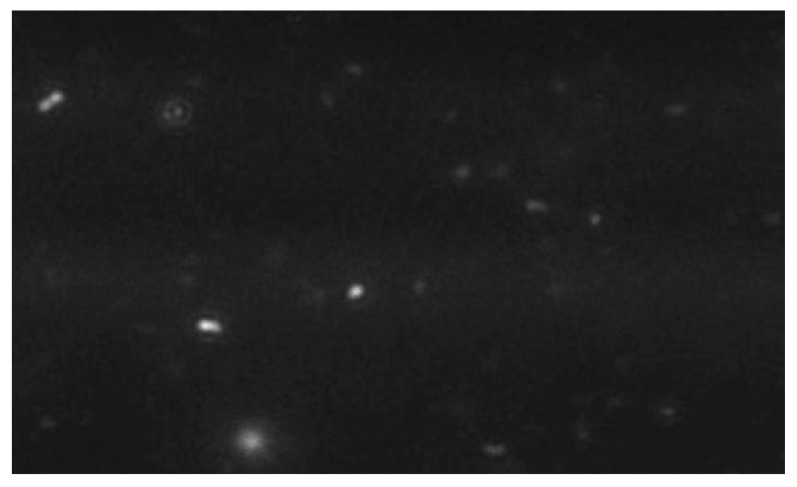
NTA tracking image of exosome preparations showing individual exosome particles.

**Figure 3 biomedicines-13-01785-f003:**
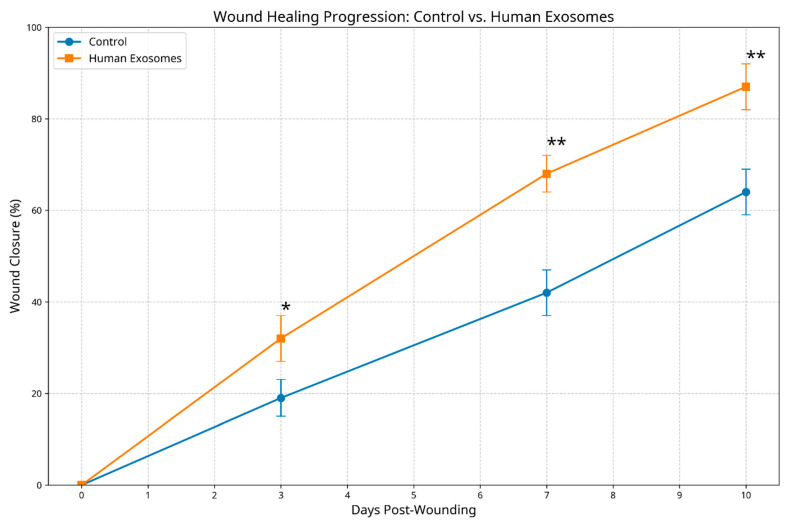
Wound closure percentage comparison between control and human exosome-treated groups over time. Exosome-treated specimens demonstrated significantly accelerated healing at all timepoints. * *p* < 0.05; ** *p* < 0.01.

**Figure 4 biomedicines-13-01785-f004:**
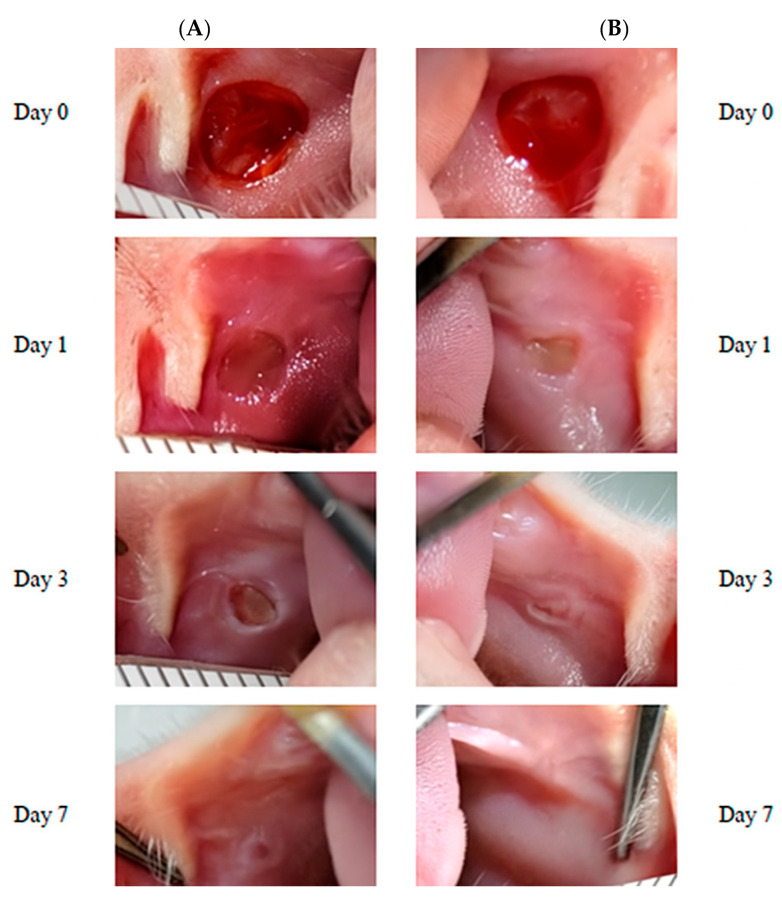
Representative macroscopic images of wound healing progression in control (**A**) and human exosome-treated (**B**) groups. Note the accelerated contraction and epithelialization in the exosome-treated specimens.

**Figure 5 biomedicines-13-01785-f005:**
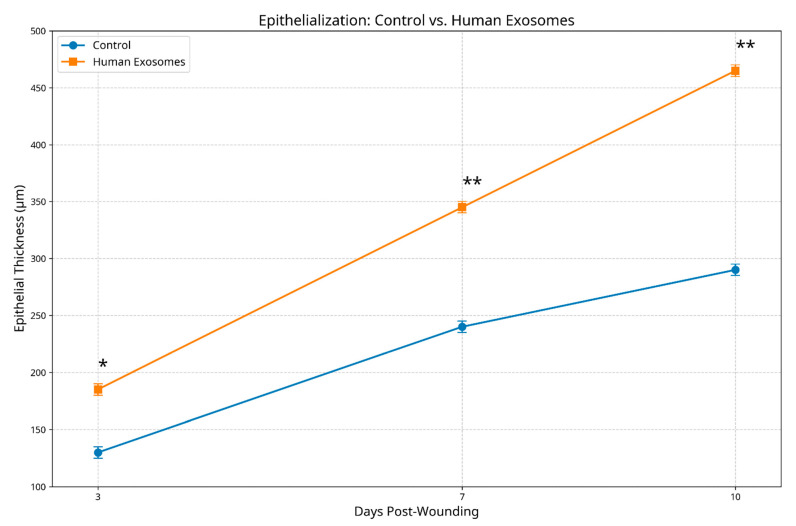
Epithelialization comparison between control and human exosome-treated groups. The graph shows epithelial thickness (μm). Exosome-treated specimens demonstrated significantly enhanced epithelial development at all timepoints. * *p* < 0.05; ** *p* < 0.01.

**Figure 6 biomedicines-13-01785-f006:**
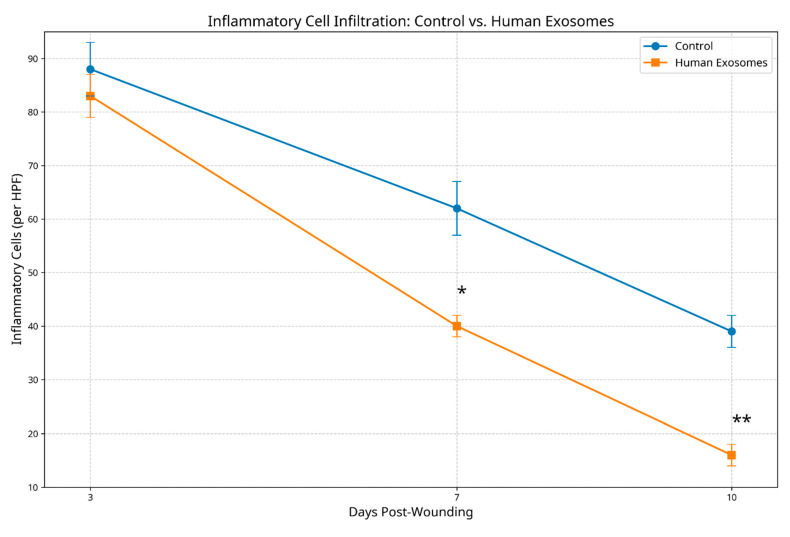
Inflammatory cell count comparison between control and human exosome-treated groups. The graph shows inflammatory cells per high-power field. Exosome-treated specimens demonstrated significantly reduced inflammatory infiltration at all timepoints. * *p* < 0.05; ** *p* < 0.01.

**Figure 7 biomedicines-13-01785-f007:**
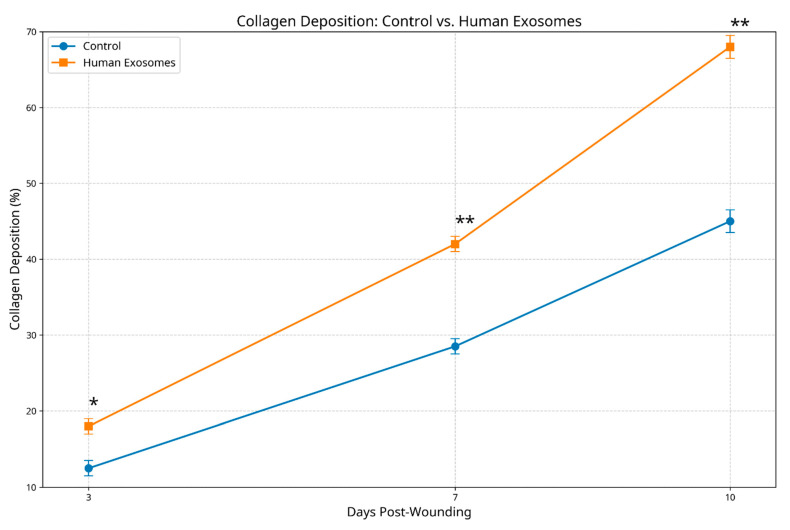
Collagen deposition comparison between control and human exosome-treated groups. The graph shows collagen density scores. Exosome-treated specimens demonstrated significantly enhanced collagen organization at all timepoints. * *p* < 0.05; ** *p* < 0.01.

**Figure 8 biomedicines-13-01785-f008:**
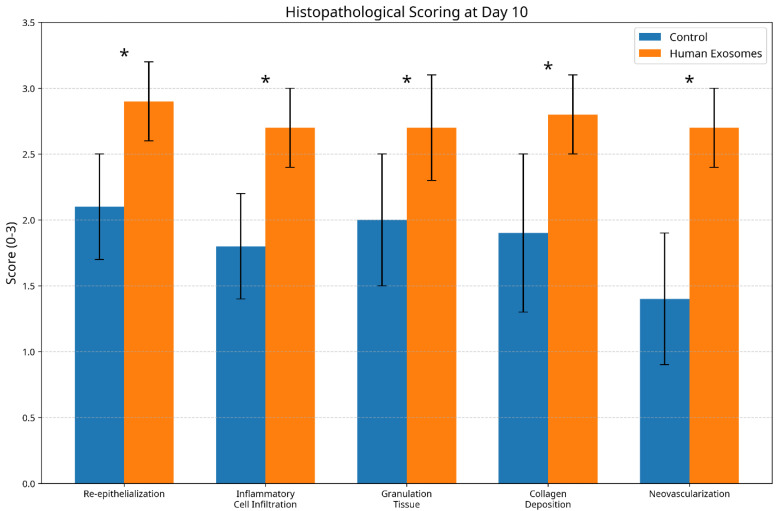
Histological parameter scores at day 10 post-wounding. The graph shows composite healing metrics comparing control and human exosome-treated groups across multiple parameters. Exosome-treated specimens demonstrated consistently superior regenerative outcomes. * *p* < 0.05.

**Figure 9 biomedicines-13-01785-f009:**
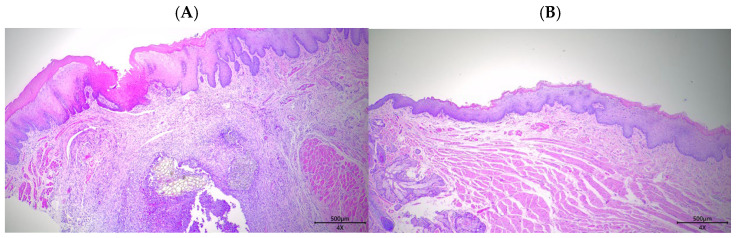
Representative histological sections at day 10 post-wounding showing (**A**) control and (**B**) exosome-treated oral mucosal tissue (H&E and Masson’s trichrome, 40×). The exosome-treated group demonstrates enhanced epithelial regeneration, reduced inflammation, and improved tissue organization.

## Data Availability

The datasets generated and/or analyzed during the current study are available from the corresponding authors on reasonable request.

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
