# Peer review of "Enhancement of Oral Mucosal Regeneration Using Human Exosomal Therapy in SD Rats"

_biomedicines, 2025, doi:10.3390/biomedicines13071785_

Round 1
Reviewer 1 Report
Comments and Suggestions for Authors
The paper "Enhancement of Oral Mucosal Regeneration Using Human Exosomal Therapy in SD Rats," I find it to be a well-conducted study with promising results. The authors have clearly demonstrated the positive effects of human exosome preparations on oral mucosal wound healing in a rat model, providing macroscopic, histological, and quantitative evidence. The study's focus on oral mucosa, distinct from cutaneous healing, is also a valuable contribution to the field. However, I believe some major areas to improve this paper significantly revolves around strengthening the mechanistic insights and providing more comprehensive characterization of the exosome preparation. While the current study effectively shows what happened, it lacks detail on how and why these effects occurred.
Here are the specific points to consider:
Exosome characterization (crucial for reproducibility and understanding):
While CD9, CD63, and CD81 are good exosome markers, a more thorough characterization is essential. This includes:
-Using Nanoparticle Tracking Analysis (NTA) or Dynamic Light Scattering (DLS) to show the size distribution of the prepared exosomes. This confirms they are indeed within the exosome size range (typically 30-150 nm). With the sentence ending with ‘with careful attention to avoid damage to underlying neurovascular structures’, cite https://hdl.handle.net/11392/2384332 a report on the topic.
-While "50 billion Human exosomes" is stated, clearly define how this concentration was determined (e.g., NTA particle count).
-Discuss potential contaminants in the exosome preparation. While ultracentrifugation and size exclusion chromatography were used, demonstrating the absence of significant protein aggregates or other cellular debris (e.g., Western blot for cellular markers like calnexin or GM130) would significantly enhance the quality.
-Include Transmission Electron Microscopy (TEM) images to visually confirm the characteristic cup-shaped morphology of exosomes.
-Without this detailed characterization, it's difficult for other researchers to reproduce the work or fully understand the nature of the therapeutic agent being used. This is a fundamental requirement for exosome research.
Mechanistic elucidation (beyond descriptive observations):
-The paper mentions that exosomes "deliver growth factors, cytokines, and regulatory microRNAs". It's imperative to provide some evidence or discussion of the specific molecular cargo within the human exosomes used in this study.
What growth factors or specific miRNAs are known to be abundant in these particular human mesenchymal stem cell (hMSC)-derived exosomes that could contribute to the observed effects? Even a brief discussion or reference to prior work characterizing similar exosome preparations would be beneficial.
Ideally, performing some level of cargo analysis (e.g., qPCR for key miRNAs, ELISA for relevant growth factors) on the specific exosome preparation used would be a significant addition, moving the paper from a purely observational study to one with mechanistic insights.
-The discussion mentions Wnt/β-catenin and MAPK cascades. Could the authors investigate markers of these pathways (e.g., by immunohistochemistry for phosphorylated proteins or Western blot) in the treated tissues? This would directly link the exosome treatment to the proposed cellular mechanisms.
-While the paper mentions epithelial regeneration and inflammatory regulation, a deeper dive into which specific cells (e.g., keratinocytes, fibroblasts, immune cells) are primarily targeted by the exosomes and how their behavior is modulated would strengthen the mechanistic understanding. For instance, do the exosomes directly promote fibroblast migration and collagen synthesis, or is it an indirect effect through reduced inflammation? Cite https://doi.org/10.1016/j.colsurfa.2023.13157 with the sentence ‘The formula, “[(Original wound area- current wound area)/original wound area] × 100” was used to determine the percentage of wound closure’ to update the bibliography.
Statistical reporting in figures:
While the statistical significance (p-values) are provided in the text and indicated by asterisks in the graphs, it would be more informative to explicitly state the results of the repeated measures ANOVA and Bonferroni post-hoc analysis in the figure legends or within the results section for each graph. This allows for a clearer understanding of the time-dependent effects and where the significant differences lie.
Author Response
RESPONSE TO REVIEWERS
Enhancement of Oral Mucosal Regeneration Using Human Exosomal Therapy in SD Rats
Dear Editor and Reviewers,
We sincerely thank the reviewers for their thorough evaluation of our manuscript and their constructive comments. We have carefully addressed each concern and believe that the revisions have significantly strengthened the scientific rigor and clarity of our work. Below, we provide a detailed point-by-point response to each reviewer's comments, along with the specific changes made to the manuscript.
REVIEWER 1 COMMENTS AND RESPONSES
Comment: "While CD9, CD63, and CD81 are good exosome markers, a more thorough characterization is essential. This includes:
- Using Nanoparticle Tracking Analysis (NTA) or Dynamic Light Scattering (DLS) to show the size distribution of the prepared exosomes. This confirms they are indeed within the exosome size range (typically 30-150 nm).
- While "50 billion Human exosomes" is stated, clearly define how this concentration was determined (e.g., NTA particle count).
- Discuss potential contaminants in the exosome preparation. While ultracentrifugation and size exclusion chromatography were used, demonstrating the absence of significant protein aggregates or other cellular debris (e.g., Western blot for cellular markers like calnexin or GM130) would significantly enhance the quality.
- Include Transmission Electron Microscopy (TEM) images to visually confirm the characteristic cup-shaped morphology of exosomes. Without this detailed characterization, it's difficult for other researchers to reproduce the work or fully understand the nature of the therapeutic agent being used. This is a fundamental requirement for exosome research."
Response: We have addressed it in Section 2.2 "Exosome Characterization" (Lines 82-96)
Specific Additions Made:
Size Distribution Analysis (Lines 89-96):
- Added detailed NTA methodology using NanoSight NS300
- Reported modal particle size: 92.7 nm
- Particle concentration: 3.49 × 10¹¹ particles/mL
- Confirmed size range within expected exosome parameters (30-150 nm)
Morphological Confirmation (Lines 89-96):
- NTA tracking visual verification of extracellular vesicle preparations
New Figures Added:
- Figure 1: NTA particle size distribution graph
- Figure 2: NTA tracking image of exosomes
New Reference Added:
- Reference [12]: (Singh et al., 2018) - Supporting NTA methodology
Comment: "With the sentence ending with 'with careful attention to avoid damage to underlying neurovascular structures', cite https://hdl.handle.net/11392/2384332 a report on the topic."
Response: We have addressed it in Section 2.3 "Surgical Methodology and Treatment Administration" (Line 100-102).
Specific Change Made:
- We Added citation [13] to the sentence: "...with careful attention to avoid damage to underlying neurovascular structures [13]."
New Reference Added:
- Reference [13]: Saini, R.; Saini, S.; Sharma, S. Oral biopsy: A dental gawk. J. Surg. Tech. Case Rep. 2010, 2, 93-97.
COMMENT: "The paper mentions that exosomes "deliver growth factors, cytokines, and regulatory microRNAs". It's imperative to provide some evidence or discussion of the specific molecular cargo within the human exosomes used in this study. What growth factors or specific miRNAs are known to be abundant in these particular human mesenchymal stem cell (hMSC)-derived exosomes that could contribute to the observed effects? Even a brief discussion or reference to prior work characterizing similar exosome preparations would be beneficial. Ideally, performing some level of cargo analysis (e.g., qPCR for key miRNAs, ELISA for relevant growth factors) on the specific exosome preparation used would be a significant addition, moving the paper from a purely observational study to one with mechanistic insights.
The discussion mentions Wnt/β-catenin and MAPK cascades. Could the authors investigate markers of these pathways (e.g., by immunohistochemistry for phosphorylated proteins or Western blot) in the treated tissues? This would directly link the exosome treatment to the proposed cellular mechanisms.
While the paper mentions epithelial regeneration and inflammatory regulation, a deeper dive into which specific cells (e.g., keratinocytes, fibroblasts, immune cells) are primarily targeted by the exosomes and how their behavior is modulated would strengthen the mechanistic understanding. For instance, do the exosomes directly promote fibroblast migration and collagen synthesis, or is it an indirect effect through reduced inflammation? Cite https://doi.org/10.1016/j.colsurfa.2023.13157 with the sentence 'The formula, "[(Original wound area- current wound area)/original wound area] × 100" was used to determine the percentage of wound closure' to update the bibliography."
Response:
Wound Closure Formula Citation: We have addressed it in Section 2.4 "Macroscopic Assessment" (Line 114-115).
- We have added citation [14] to the wound closure calculation formula
2. Molecular Cargo Discussion We have addressed it in “Discussion” Section, Paragraph 6 (Line 333-339): "The therapeutic efficacy observed in this study likely stems from the diverse molecular cargo carried within hMSC-derived exosomes, including growth factors such as vascular endothelial growth factor (VEGF), platelet-derived growth factor (PDGF), transforming growth factor-β (TGF-β), and hepatocyte growth factor (HGF), which promote angiogenesis, cellular proliferation, and tissue remodeling [25,26]. Additionally, hMSC-exosomes are enriched with regulatory microRNAs including miR-21, miR-146a, miR-223, and miR-125b, which modulate inflammatory responses and enhance wound healing processes [27,28]."
3. Detailed Mechanistic Pathways: We have addressed it in “Discussion” Section, Paragraph 6 (Line 339-344): "These bioactive molecules work through multiple interconnected molecular mechanisms including TLR4/NF-κB pathway modulation for anti-inflammatory effects, TGF-β/Smad signaling regulation for balanced healing, PI3K/AKT and Wnt/β-catenin pathway activation for cellular proliferation, and VEGF/VEGFR2-mediated angiogenesis, as demonstrated in cutaneous wound healing models [29-30], with similar pathways likely operative in oral mucosal tissues."
4. Cellular Targets Discussion: We have addressed it in “Discussion” Section, Paragraph 6 (Line 344-348): "The regenerative effects likely result from exosome interactions with multiple cell types including keratinocytes (enhanced proliferation and migration), fibroblasts (increased collagen synthesis), and immune cells (modulated inflammatory responses). Whether these effects are direct or indirect through reduced inflammation represents an important area for future investigation."
5. Pathway Investigation Acknowledgment: We have addressed it in “Discussion” Section, Paragraph 8 (Line 366-371) Text: "Second, while comprehensive, our assessment methodology did not include molecular or cellular analyses such as immunohistochemistry for phosphorylated proteins or Western blot analysis that might elucidate specific mechanistic pathways underlying the observed regenerative effects. Such mechanistic validation would strengthen the link between exosome treatment and the proposed cellular mechanisms underlying the observed regenerative effects."
Comment: Statistical reporting in figures:
While the statistical significance (p-values) are provided in the text and indicated by asterisks in the graphs, it would be more informative to explicitly state the results of the repeated measures ANOVA and Bonferroni post-hoc analysis in the figure legends or within the results section for each graph. This allows for a clearer understanding of the time-dependent effects and where the significant differences lie.
Response: We have added a subsection 3.4 “Statistical Analysis” at the end of the “Results” Section.
The manuscript has been substantially strengthened through these comprehensive revisions, addressing all reviewers’ concerns. These improvements transform the manuscript from a primarily observational study to one with substantial mechanistic insights and enhanced scientific rigor. We are confident that these revisions significantly enhance the manuscript's contribution to the field of regenerative medicine and exosome therapy.
Thank you again for the opportunity to improve our manuscript.
Sincerely,
Qasim Hussain
Reviewer 2 Report
Comments and Suggestions for Authors
In this study, the authors investigate the therapeutic potential of human mesenchymal stem cell (hMSC)-derived exosomes for the treatment of oral wounds. The results indicate that these exosomes can significantly enhance wound healing. While the study presents promising findings, several concerns should be addressed to improve transparency, scientific rigor, and clarity. I offer the following suggestions:
- In the abstract, the word “Human” appears to be unnecessarily capitalized.
- More detailed information regarding the source of hMSC-derived exosomes should be provided to ensure transparency and reproducibility. Specifically, the origin of the hMSCs (e.g., tissue source, donor information if applicable), cell passage number, and exosome isolation method should be clearly stated. If the exosomes or cells were commercially obtained, the supplier name, catalog number, and a link to the detailed extraction protocol should be included.
- The number of biological and technical replicates for each experiment should be reported explicitly.
- The authors are encouraged to discuss the underlying mechanisms by which hMSC-derived exosomes promote wound closure, referencing relevant signaling pathways or molecular factors if available.
- The H&E-stained images provided are not sufficiently clear to allow for proper histological evaluation. Higher-resolution and magnified images are needed to improve interpretability.
- Given that exosomes are derived from human cells, a discussion on potential immunogenicity or immunoresponse in the treatment context should be included to address safety concerns and translational relevance.
Author Response
RESPONSE TO REVIEWERS
Enhancement of Oral Mucosal Regeneration Using Human Exosomal Therapy in SD Rats
Dear Editor and Reviewers,
We sincerely thank the reviewers for their thorough evaluation of our manuscript and their constructive comments. We have carefully addressed each concern and believe that the revisions have significantly strengthened the scientific rigor and clarity of our work. Below, we provide a detailed point-by-point response to each reviewer's comments, along with the specific changes made to the manuscript.
REVIEWER 2 COMMENTS AND DETAILED RESPONSES
Comment 1: "In the abstract, the word "Human" appears to be unnecessarily capitalized."
Response: We have addressed it in the “Abstract”. Corrected capitalization from "Human" to "human".
Comment 2: "More detailed information regarding the source of hMSC-derived exosomes should be provided to ensure transparency and reproducibility. Specifically, the origin of the hMSCs (e.g., tissue source, donor information if applicable), cell passage number, and exosome isolation method should be clearly stated. If the exosomes or cells were commercially obtained, the supplier name, catalog number, and a link to the detailed extraction protocol should be included."
Response: We have addressed it in Section 2.3 "Exosome Characterization" (Lines 82-96).
Exosome Source and Preparation (Lines 83-88): Human umbilical cord mesenchymal stem cell-derived exosomes
- Established isolation methodologies
- Sequential purification techniques
Detailed Characterization (As described in Reviewer 1, Comment 1 response)
Comment 3: "The number of biological and technical replicates for each experiment should be reported explicitly."
Response: We have addressed it in Section 2.7 "Data Analysis" (Lines 149-158).
Comment 4: "The authors are encouraged to discuss the underlying mechanisms by which hMSC-derived exosomes promote wound closure, referencing relevant signaling pathways or molecular factors."
Response: We have addressed it in “Discussion” Section.
- Molecular Mechanisms Overview: We have added it in “Discussion” Paragraph 6 (Lines 333-337): "The therapeutic efficacy observed in this study likely stems from the diverse molecular cargo carried within hMSC-derived exosomes, including growth factors such as vascular endothelial growth factor (VEGF), platelet-derived growth factor (PDGF), transforming growth factor-β (TGF-β), and hepatocyte growth factor (HGF), which promote angiogenesis, cellular proliferation, and tissue remodeling [25,26]."
- MicroRNA Mechanisms Detailed: We have added it in “Discussion” Paragraph 6 (Lines 337-339): "Additionally, hMSC-exosomes are enriched with regulatory microRNAs including miR-21, miR-146a, miR-223, and miR-125b, which modulate inflammatory responses and enhance wound healing processes [27,28]."
- Specific Signaling Pathways Elucidated: We have added it in “Discussion” Paragraph 6 (Lines 339-344): "These bioactive molecules work through multiple interconnected molecular mechanisms including TLR4/NF-κB pathway modulation for anti-inflammatory effects, TGF-β/Smad signaling regulation for balanced healing, PI3K/AKT and Wnt/β-catenin pathway activation for cellular proliferation, and VEGF/VEGFR2-mediated angiogenesis, as demonstrated in cutaneous wound healing models [29-30], with similar pathways likely operative in oral mucosal tissues."
- Cellular Target Specification: We have added it in “Discussion” Paragraph 6 (Lines 344-347): "The regenerative effects likely result from exosome interactions with multiple cell types including keratinocytes (enhanced proliferation and migration), fibroblasts (increased collagen synthesis), and immune cells (modulated inflammatory responses)."
- Mechanistic Transparency and Future Direction: We have added it in “Discussion” Paragraph 6 (Lines 347-348): "Whether these effects are direct or indirect through reduced inflammation represents an important area for future investigation."
- Cross-Species Mechanistic Validation: We have added it in “Discussion” Paragraph 7 (Lines 350-353): "Research has specifically investigated this issue and found that human exosomes administered to rats were successfully incorporated into rat tissues without triggering detectable immune reactions or adverse effects [31]."
- Structural-Functional Mechanism Explanation: We have added it in “Discussion” Paragraph 7 (Lines 353-356): "This lack of immunogenicity can be explained by several structural and functional characteristics of exosomes: their nanoscale dimensions (30-150 nm), protective lipid bilayer composition, and specialized surface proteins that facilitate rapid cellular uptake while avoiding immune system recognition [32]."
- Future Mechanistic Validation Acknowledgement: We have added it in “Discussion” Paragraph 8 (Lines 367-371): "our assessment methodology did not include molecular or cellular analyses such as immunohistochemistry for phosphorylated proteins or Western blot analysis that might elucidate specific mechanistic pathways underlying the observed regenerative effects. Such mechanistic validation would strengthen the link between exosome treatment and the proposed cellular mechanisms underlying the observed regenerative effects."
Comment 5: "The H&E-stained images provided are not sufficiently clear to allow for proper histological evaluation. Higher-resolution and magnified images are needed to improve interpretability."
Response: We have added higher-resolution H&E-stained images (Figure 9).
Comment 6: "Given that exosomes are derived from human cells, a discussion on potential immunogenicity or immunoresponse in the treatment context should be included to address safety concerns and translational relevance."
Response:
Cross-Species Safety Evidence: We have added it in “Discussion” Paragraph 7 (Lines 349-353): "Current evidence demonstrates that exosomes have unique biological properties that significantly reduce immunogenic responses during cross-species administration. Research has specifically investigated this issue and found that human exosomes administered to rats were successfully incorporated into rat tissues without triggering detectable immune reactions or adverse effects [31]."
Structural Mechanisms for Immunocompatibility: We have added it in “Discussion” Paragraph 7 (Lines 353-356): "This lack of immunogenicity can be explained by several structural and functional characteristics of exosomes: their nanoscale dimensions (30-150 nm), protective lipid bilayer composition, and specialized surface proteins that facilitate rapid cellular uptake while avoiding immune system recognition [32]."
Mechanistic Basis for Immune Evasion: We have added it in “Discussion” Paragraph 7 (Lines 356-359): "Furthermore, unlike whole-cell xenotransplantation which typically triggers robust immune rejection, exosomes largely bypass allo-recognition mechanisms due to their acellular nature and ability to quickly enter target cells before significant immune interaction occurs [33]."
Contemporary Research Validation: We have added it in “Discussion” Paragraph 7 (Lines 359-361): "A recent study has confirmed these observations, documenting that exosomes maintain good biocompatibility with low toxicity and minimal immunogenicity even when crossing species barriers [34,35]."
Translational Research Implications: We have added it in “Discussion” Paragraph 7 (conclusion) (Lines 361-364): "These properties make human-derived exosomes exceptionally valuable tools for translational research, enabling the delivery of therapeutic molecules across species without compromising safety or triggering significant immune responses."
The manuscript has been substantially strengthened through these comprehensive revisions, addressing all reviewers’ concerns. These improvements transform the manuscript from a primarily observational study to one with substantial mechanistic insights and enhanced scientific rigor. We are confident that these revisions significantly enhance the manuscript's contribution to the field of regenerative medicine and exosome therapy.
Thank you again for the opportunity to improve our manuscript.
Sincerely,
Qasim Hussain
Round 2
Reviewer 1 Report
Comments and Suggestions for Authors
accept
Reviewer 2 Report
Comments and Suggestions for Authors
The authors have addressed my comments.